# General-purpose programmable photonic processor for advanced radiofrequency applications

Daniel Pérez-López [1,2] ✉, Ana Gutierrez [1,2], David Sánchez [2], Aitor López-Hernández [1], Mikel Gutierrez[2], Erica Sánchez-Gomáriz [1,2], Juan Fernández [2], Alejandro Cruz [2], Alberto Quirós[2], Zhenyun Xie [2], Jesús Benitez [2], Nandor Bekesi [2], Alejandro Santomé [2], Diego Pérez-Galacho[1], Prometheus DasMahapatra[1], Andrés Macho [1] & José Capmany [1,2] ✉

A general-purpose photonic processor can be built integrating a silicon photonic programmable core in a technology stack comprising an electronic monitoring and controlling layer and a software layer for resource control and programming. This processor can leverage the unique properties of photonics in terms of ultra-high bandwidth, high-speed operation, and low power consumption while operating in a complementary and synergistic way with electronic processors. These features are key in applications such as next-generation 5/6 G wireless systems where reconfigurable filtering, frequency conversion, arbitrary waveform generation, and beamforming are currently provided by microwave photonic subsystems that cannot be scaled down. Here we report the first general-purpose programmable processor with the remarkable capability to implement all the required basic functionalities of a microwave photonic system by suitable programming of its resources. The processor is fabricated in silicon photonics and incorporates the full photonic/electronic and software stack.

Programmable photonic circuits manipulate the flow of light on a chip by electrically controlling a set of tunable analog gates connected by optical waveguides[1–4]. Light is distributed and spatially rerouted to implement various linear functions by interfering signals along different paths. The proliferation of ultra-high-speed 5/6 G mobile networks[5,6], satellite communications[7,8], photonic computing[8], advanced communications[9], lidar[10], microwave spectroscopy[11], edge cloud computing[12,] and the internet of things[13] is exerting considerable pressure over existing hardware infrastructures connecting the wireless and fiber segments in communication networks. The solution to accommodating the ever-increasing demands for bandwidth, providing lower latency and reduced power consumption passes through extending the radiofrequency operation spectrum to the microwave

and millimeter-wave regions and developing compact, flexible, and agile solutions capable of interfacing the radiofrequency (RF) and photonic domains. Microwave photonics[14,15] (MWP), which uses optical devices and techniques to generate, manipulate, transport, and measure high-speed radiofrequency signals is one of the few technologies capable of supporting this evolution. Traditional RF systems and devices that process signals directly at frequencies $f_{RF}$ in the RF spectrum are static, bulky, vulnerable to electromagnetic interference (EMI), and have limited frequency bandwidth (see supplementary note 1). The translation of RF systems into the optical region of the spectrum using frequency up-conversion to the optical spectrum region $f_o$ (see supplementary note 1) brings the possibility of leveraging the advantages of photonic systems, which include tunability,

[1]Photonics Research Labs, iTEAM Research Institute, Universitat Politècnica de València, Valencia, Spain. [2]iPronics, Programmable Photonics, Valencia, Spain. ✉e-mail: daniel.perez@ipronics.com; jcapmany@iteam.upv.es

broadband operation, immunity to EMI, and potential space weight and power (SWAP) gains. Most of these advantages have been proved in prior works where discrete photonic components such as lasers, optical fibers, filters, and detectors have been assembled to implement specific functionalities such as waveform generation[16], beamforming[17–19], filtering[20–24], channelization[25], and instantaneous frequency measurements[26]. Though interesting, these solutions are limited in terms of reliability, and repeatability and are still bulky and complex to engineer and operate. Therefore, their applicability to scenarios with massive takeovers and high volumes is severely hampered.

Many of the former limitations can be overcome by benefiting from the compact footprint, modularity, and scalable fabrication methods of integrated photonic circuits[27–31]. The convergence of the two fields, known as integrated MWP[5,32,33] has allowed a dramatic reduction in the footprint and losses of complex MWP systems allowing a limited degree of reconfigurability. However, most of the integrated photonic microwave subsystems reported to date have been implemented as application-specific photonic integrated circuits (ASPICs), which are designed to perform a particular functionality[34–38]. These chips usually require several fabrication cycles to achieve optimum performance, and this results in unacceptable long fabrication times and costs[4]. Furthermore, they are very sensitive to non-recurring engineering costs (NRE) as any change in systems specifications requires a new chip design. A way out is to leverage the strong push toward programmable photonic circuits that are also being developed for related application areas such as quantum photonics[39–41], artificial intelligence[42,43], neuromorphic computing[44–46,] and sensing[47]. In this context, two particular routes are being explored. In the first, circuits based on traditional interferometric and photonic waveguide structures are designed for flexible programming of its relevant parameters. The key elements are unit cells of Mach–Zehnder interferometers and ring resonators that can be activated individually to implement, reconfigurable delay lines[48], beamformers[49], waveform generators[50], and filters[51]. The second is based on the possibility of implementing a generic signal processor or field programmable photonic gate array (FPPGA)[2,3] where a photonic core made from a mesh of uniform tunable building blocks and surrounded by external high-performance blocks can be easily programmed to support multiple functions[52–54] by software definition. The successful development of this approach requires solving an extremely challenging and complex technology stack problem[1,3,4] comprising not only the photonic processing layer but also an electronics layer providing driving, monitoring, and control, and a software layer for optimization and programming. While to the best of our knowledge, this has not been demonstrated so far, the gains that could be achieved in the cases where programmable photonic circuits can replace electronic subsystems are huge, especially in terms of ultra-high bandwidth, high-speed operation, and low power consumption. These are complemented by their flexibility and reduced fabrication costs from leveraging economies of scale. All these will be obtained while operating in a complementary and synergistic way with electronic processors. These gains will have a strong impact in meeting and surpassing the stringent requirements of mobile-based applications as they will enable a new generation of compact, broadband, and programmable devices ready to be allocated both in central and base stations.

Here we report the first general-purpose scalable programmable photonic processor with the remarkable capability to implement all the main functionalities required in a microwave photonic system by suitable programming of its resources. So far, individual microwave photonic applications are mostly carried by ASPICs, but as anticipated[1], all of them can be implemented by the same general-purpose programmable processor if it integrates the required technology stack (photonic, electronic, and software). This evolution beyond the current state of the art and the demonstration of

multifunctionality by programming are the fundamental results of the paper. The processor is fabricated in a silicon photonics platform and incorporates for the first time to our knowledge the full photonic/electronic and software stack. We leverage recent advances in advanced waveguide mesh designs and optimization algorithms[52,53], to enable software-defined functionality programing at reconfiguration speeds of several microseconds. With the proper development of attached modulators and photodetectors, the operation bandwidth of this processor can surpass the millimeter-wave band featuring power consumption values in the order of a few watts. The possibility of implementing all the functionalities with a single chip opens the path to scale down the processor size to dimensions compatible with the requirements of next-generation millimeter-wave base stations and satellites[6,7]. Our findings indicate that this processor can work in frequency ranges of up to 100 GHz featuring power consumption values in the order of a few watts. The possibility of implementing all the functionalities with a single chip opens the path for use in an unconstrained number of custom applications.

## Results
### Overview of RF-photonic functionalities
RF-photonics can support all the main functionalities needed by current and future application scenarios in communications, computing, and sensing. Figure 1 shows a selection of some of them covering terrestrial, space, and airborne scenarios with very different requirements in terms of operation frequency, power consumption, and footprint. It is precisely the added value of reconfigurability that enables this unique feature of spanning this variety of requirements as well as adaptative-demanding applications. A list of the main 12 functionalities required in these systems with a description and hosting locations is provided in Table 1. The table also provides an indicative limited selection of publications reporting their ASPIC implementation. The interested can find an exhaustive list of references reporting ASPIC implementations of MWP functionalities elsewhere[11].

### The general-purpose photonic processor
The general-purpose photonic processor presented in this work aggregates, for the first time, the complex full-stack necessary to operate a programmable photonic device: the optical layer, the control layer, and the software layer (Fig. 2a). The photonic stage is integrated into a silicon-on-insulator chip that includes a reconfigurable core of 72 Programmable Unit Cells (PUC) distributed in a flatted hexagonal mesh topology[55] (Fig. 2b). In addition, the chip includes an optoelectronic monitoring unit array that provides feedback on the optical power at each port and four high-performance filters. The chip is connected optically through a fiber array with 64 ports (Fig. 2c), from where 28 are routed to the mesh core and electronically through a wire bonding interconnection to a Printed Circuit Board (PCB).

On the optical layer, the chip is optimized for C-band operation. As described in Fig. 2b and shown in Fig. 2d, the mesh core has 40 output ports, 12 connected to on-chip high-performance blocks (2 lattice filters of order 4, 1 coupled-ring filter, and 1 Ring-assisted MZI filter of order 4). As described before, each PUC consists of a Mach-Zehnder Interferometer (MZI) with two thermo-optic phase actuators. By tuning one of the arms, the user can modify the coupling factor of the 2 × 2 block. For independent coupling factor and phase response configuration, the user/system can configure both arms[56]. The insertion loss and efficiency have been characterized across many chips and two wafers, resulting in 0.48 dB/PUC and 1.3 mW/$\pi$, respectively. The length and basic delay unit are also characterized as 811 um and 11.25 ps. Propagation losses are measured between 1.5 and 2.5 dB/cm for different waveguide widths and dies. Finally, fiber-chip coupling loss ranges between 1.5 dB to 3 dB, for different coupling techniques. For the scope of this paper, we employed a device with 3 dB loss per facet.

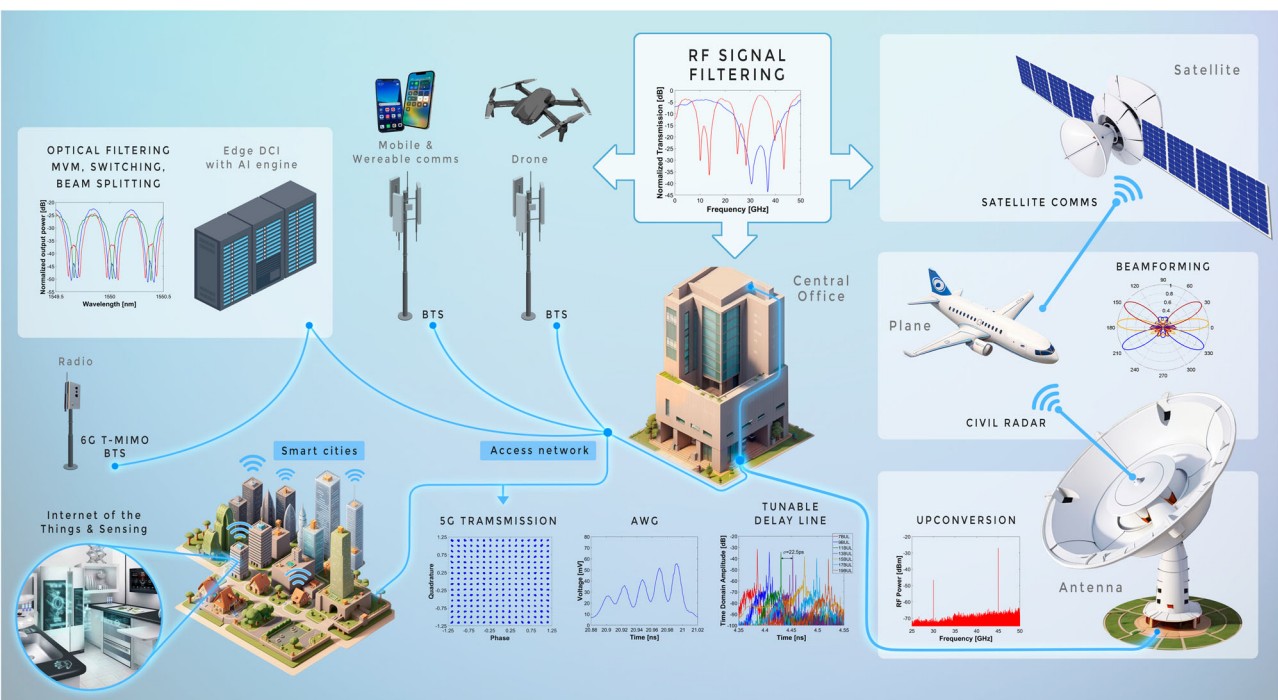

**Fig. 1 | A selection of RF-photonic functionalities covering terrestrial, space, and airborne scenarios.** (BTS base station, CS central station, DC data center, AI artificial intelligence, AWG arbitrary waveform generation, IoT Internet of things, MIMO multiple input multiple outputs).

**Table 1 | The main functionalities required in RF-photonic systems with a brief description of each one, their main applications, and where the systems implementing them are hosted**

| Functionality and some reported ASPICs | Short description | Hosting | Application |
|---|---|---|---|
| Tunable RF-filtering[5, 11,33–35,51,52] | Channel selection in the RF domain & Noise reduction | Base stations, Central offices, planes, satellites, drones, edge data centers | *Mobile & satellite comms, IoT, Sensing, Radar, Imaging* |
| Tunable optical filtering[4] | Superchannel RF selection in the optical domain & Noise reduction | | |
| Tunable RF delay line[11,48,49] | Selective RF channel delay and time-delayed signal combination | Radio over fiber links, beamformers | *Phased array antennas, Radar, Sensing, Lidar* |
| Tunable RF phase shifter[11,48] | Selective RF channel dephasing and phase-delayed signal combination | | |
| Frequency measurement[21] | Identification of the frequency of non-desired interfering signals | Radar and antenna infrastructure sensors, satellites, base stations | *Radar, navigation systems, self-interference cancellation* |
| Optoelectronic oscillators (OEOs)[11,36,37] | Ultrapure tunable RF tone generation based on combined optical/RF energy storage in a mixed optoelectronic cavity. | Clock reference systems, Spectrum analyzers, RF coherent receivers | *Avionics and Satellite comms, Navigation systems. Time and frequency measurement* |
| Arbitrary waveform generation (AWG)[11,50] | Generation of RF wavepackets with arbitrary amplitude shapes and central frequencies of operation. | Central stations, Radar, satellites, planes, drones, infrastructures, IoT terminals | *Radar, Mobile communications, microwave imaging, instrumentation* |
| Optical generation of MM-wave CW signals[11,5] | Baseband and Intermediate frequency conversion to/from the radio spectrum | Central stations, radar antennas, base stations | *Mobile, wearable & satellite comms, Radar, Imaging* |
| Beamforming[5,11,48,49] | Broadband angular steering of a radiation pattern from/to an antenna array | Base stations, Radar infrastructures | *Phased array antennas, 5-6 G mobile & satellite comms, Sensing, Radar, Imaging* |
| Radio over fiber transmission[5] | Broadband communication transceivers between central stations and base stations | Edge Data centers, Base stations, Central stations | *Mobile and satellite communications* |
| Tunable beam splitting[1,2] | Signal distribution among different physical outputs | Central stations, Edge Data Centers, base stations, satellites, planes | *Mobile &satellite communications,* |
| Tunable RF interconnection and switching[1,2] | RF Channel and superchannel connection between selected input/output ports | | |

The left column also includes a brief list of references corresponding to ASPICs reported for each of them.

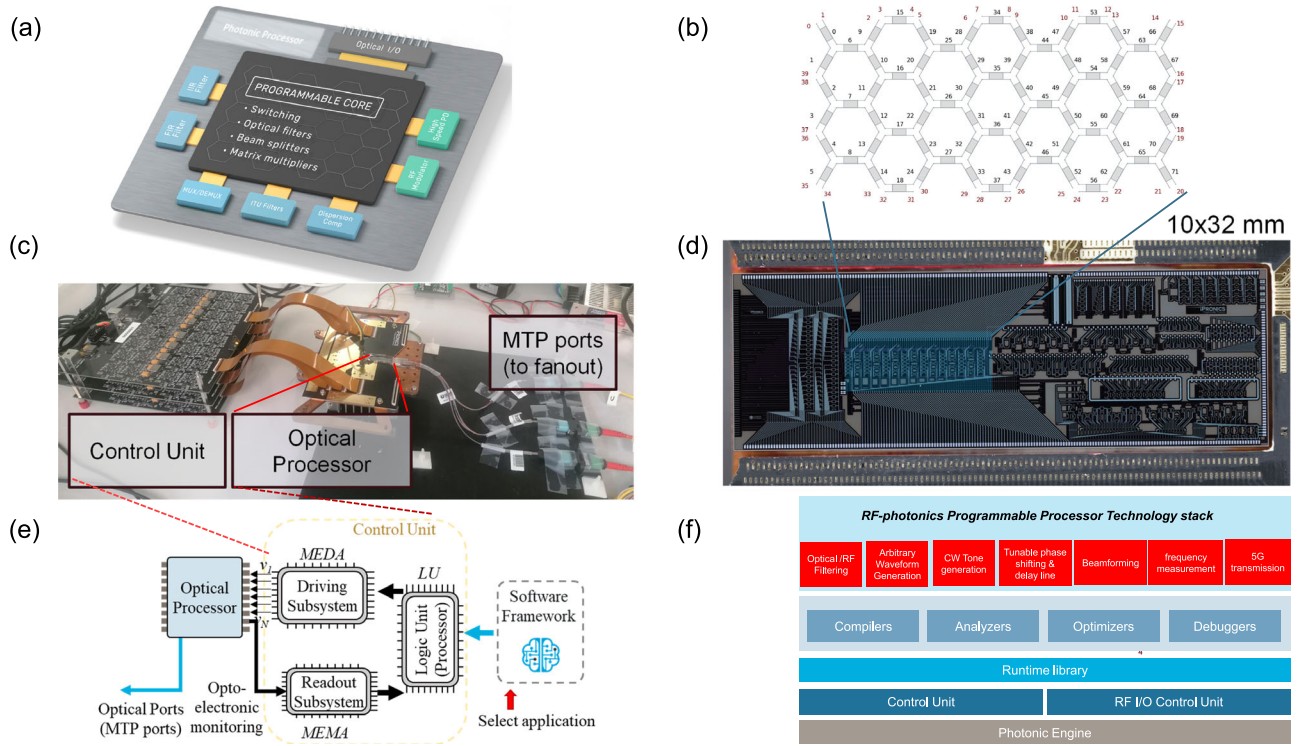

**Fig. 2 | The photonic processor elements and layers. a** Optical layer of the processor with the core, I/Os, and high-performance blocks, **b** schematic of the waveguide mesh core, **c** assembled chip with control unit and access fibers, **d** picture of the chip with the highlighted region of the reconfigurable core, **e** interconnection diagram between optical system, control unit and software layer, **f** Software layer stack employed in this work.

On the electronic layer, 304 on-chip phase actuators are controlled by a board-integrated programmable current array source connected to the Logic Unit (LU) (Fig. 2c, e). Similarly, 40 on-chip photodetectors are measured through an on-board readout system connected to the logic unit. Closing the workflow, the software running in the LU actuates over the photonic system and can get instant data of the circuit configuration. The overall operation (set and readout times) is dominated by the driving unit, setting the reconfiguration time of the system to 15–90 ms.

Finally, the software layer includes the backend functions necessary to maintain the chip temperature stable, drive and read from the photonic electro-optic components, and the application layers that automate the configuration of applications at different abstraction levels. Additional information on the software layer can be obtained in Supplementary Note 2.

## Experiments

We employed the general-purpose photonic processor to demonstrate all the 12 main MWP functionalities listed in Table 1. Prior to any experimental demonstration we performed a loss and delay characterization of the photonic core (see details in supplementary note 3), which rendered a loss per PUC (coupling + propagation + bend) of 0.48 dB ± 0.02 dB, a time delay per PUC of 11.2 ps and a chip excess loss (fiber-to-chip + access loss) of 10.5 dB. While here we outline only the salient results the reader can find these, and others described in more detail in the supplementary notes 3 to 11 inclusively.

**Tunable and reconfigurable photonic filters.** Programmable photonic filters are the key element required for the implementation of most of the functionalities needed in MWP systems (see Supplementary Note 0). Between the electro-optic and optoelectronic interfaces, the reconfigurable optical core will include one or more optical filters and exploit either their spectral or time-domain responses depending on the application. It is therefore essential to

have enough flexibility to implement a wide range of filter families. In our processor we can program and implement both finite (FIR) as well as infinite (IIR) impulse response reconfigurable photonic filters including unbalanced Mach-Zehnder Interferometers (MZIs) or UMZIs, Optical Ring Resonators (ORRs), Ring-assisted Mach-Zehnder Interferometer filters (RAMZIs), UMZI lattice filters, Coupled Resonant Waveguide filters (CROWs), Side Coupled Integrated Spaced Sequence of Optical Resonators (SCISSORs), Single and double loaded RAMZI filters and simultaneous UMZI, RR, and RAMZI filters. Figure 3a shows the schematic of the waveguide mesh implemented by its photonic core including the I/O port and PUC numbering and programmed to implement a resonant third-order CROW filter featuring three coupled-ring cavities of length 6 Basic Unit Length (BUL) (see attached circuit layout). This filter has two complementary outputs representing the reflection and transmission of a resonant filter. Figure 3b shows the spectral response for the reflection spectrum. The filter can be optimized by tuning the internal PUC responses and changing the zero and pole positions[57]. For the particular case shown in the left part of Fig. 3b the filter yields a notch-type response with 26 dB suppression and a free spectral range (FSR) of 14.98 GHz. The right part of Fig. 3b shows the frequency notch tuning along a complete FSR (using an intracavity PUC as a cross-phase (CP) shifter). The transmission spectrum is shown in the left part of Fig. 3c. As expected, a complementary bandpass characteristic is obtained, now with an extinction ratio of 23 dB. The right part of 3(c) shows the bandpass tuning along a complete FSR (using an intracavity PUC as a cross-phase shifter). Figures 3d, e show the programming and the measurement and tuning results for a non-resonant or IIR filter, in this case, a third-order UMZI lattice filter with 2BUL path imbalance. The filter features a bandpass characteristic with an extinction ratio of 21 dB and an FSR of 44.95 GHz. Again, the bandpass position can be fully tuned along an FSR interval by operating a cross-phase shifter in one of the interferometer arms. In Supplementary Note 4 we provide a detailed description of the

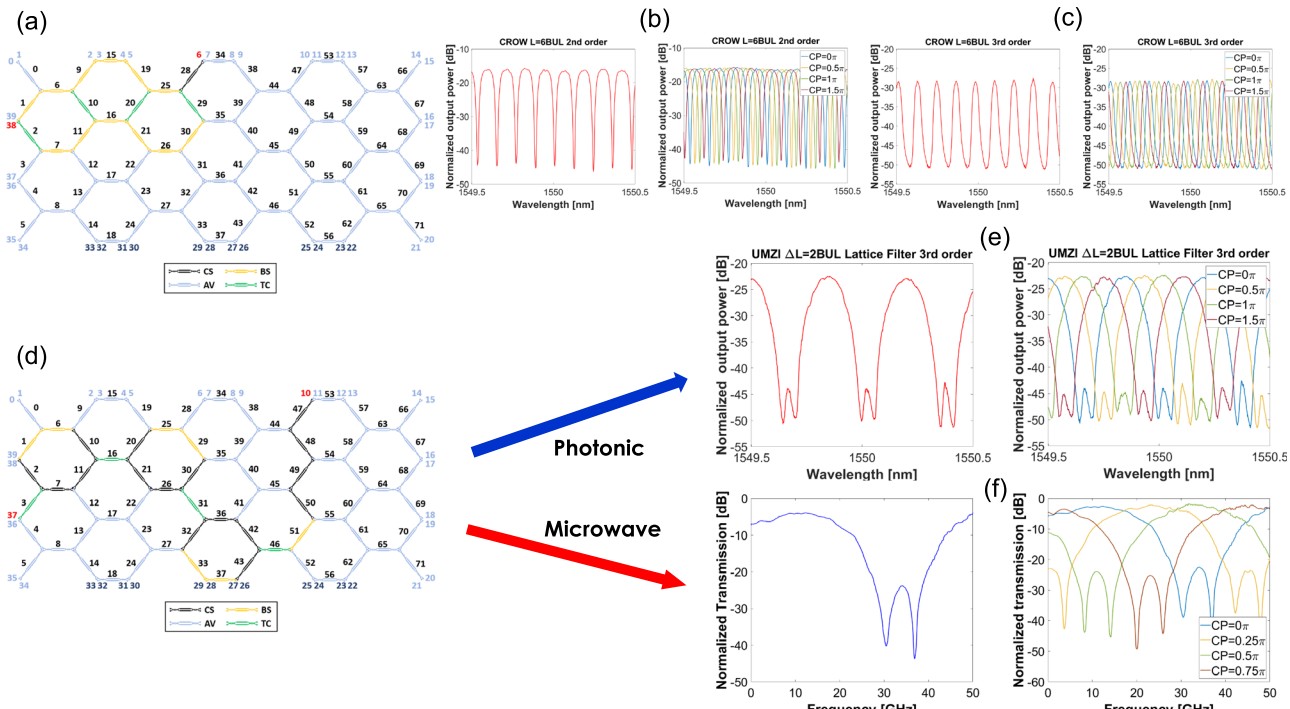

**Fig. 3 | Measured results for Photonic and RF-photonic filtering and phase shifting. a** Schematic of the waveguide mesh implemented by its photonic core including the I/O port and PUC numbering and programmed to implement a resonant (IIR) third-order CROW filter featuring three coupled-ring cavities of length 6BUL (see attached circuit layout). PUC programming color code (CS Cross State switch, BS: Bar State switch, TC: Tunable Coupler, AV: available). **b** Left: Spectral response for the reflection spectrum. **b** Right: frequency notch tuning along a complete FSR (using an intracavity PUC as a cross-phase shifter CP). **c** Left: Transmission spectrum, **c** Right: bandpass tuning along a complete FSR (using an intracavity PUC as a cross-phase shifter CP). **d** Schematic of the waveguide mesh programmed to implement a third-order UMZI photonic lattice filter. **e** Optical transfer function and the tuning of the UMZI lattice filter and **f** RF transfer function and the tuning of the MWP filter based on self-beating of the third-order UMZI photonic lattice filter. PUC programming color code (CS Cross State switch, BS: Bar State switch, TC: Tunable Coupler, AV: available).

implementation and measurements of 17 additional types of filter architectures.

**Tunable and reconfigurable radiofrequency filters and phase shifters.** Programmable radiofrequency filters can be obtained from direct down-conversion of the spectrum of an optical filter. For this, an input single-sideband RF signal must be employed, the optical carrier must be suppressed and reinjected, and combined with the upconverted RF sideband after the latter is processed by the filter. We assembled a testing and measurement setup as described in Supplementary Note 5 to measure the implementation of MWP filters based on different photonic filters. Figure 3e, f shows as an example, the transfer function and the tuning of an MWP filter based on the third-order UMZI photonic lattice filter described in the prior subsection. Note that the filter displays an RF power extinction ratio of 40 dB and features a radio-frequency FSR of 44 GHz. The interested reader is directed to Supplementary Note 5 for the implementation details of this and another MWP filter based on the down-conversion of a resonant RAMZI optical filter.

**Optical generation of millimeter-wave CW signals.** Millimeter-wave signals can be generated by optical means through a variety of techniques. A flexible approach leverages the frequency up-conversion capability of an external modulator when a continuous wave signal provided by a laser is modulated by an RF tone followed by a tunable photonic filter to select the desired high-order harmonic sideband at its output. Up-conversion is produced by the beating of the optical carrier and the selected harmonic sideband at the photodetector. Figure 4a shows the assembled experimental configuration using the programmable photonics processor (see details in Supplementary Note 6). The light from a tunable laser is split into two paths through a 1 × 2 optical coupler (OC). On one path, the light is delivered to an intensity modulator (MOD) driven by a signal at the RF input frequency (15 GHz) to be multiplied. A tunable optical filter removes both the optical carrier and the lower sidebands at the modulator output. After an amplification stage, a programmable optical integrated filter is synthesized in the photonic processor, to select a single harmonic tone and ensure the suppression of all the remaining spectral components. Finally, a second optical coupler (OC) recombines the isolated tone with the optical carrier coming from the second output port of the input OC, and the resulting signal is optically amplified and delivered to a high-speed photodiode (PD). At the PD output, an RF carrier wave is generated at the beating frequency given by the detuning between the optical carrier and the selected modulator harmonic. Tuning the synthesized filter in the mesh by sweeping the cross phases of some PUCs in the filter path for selecting different tones allows the synthesizing of RF signals at integer multiples of the original RF frequency (N x 15 GHz). Figure 4b, c shows the x2 (30 GHz) and x3 (45 GHz) obtained when the third-order UMZI lattice filter shown in Fig. 3d, e is used as a discriminating filter. Here the resonance tuning capability provided by the cross-phase filters is exploited (see details in Supplementary Note 6) to select the second and the third-order harmonics, respectively, yielding electrical rejections of 22 dB for N = 2 and 19.5 dB for N = 3, respectively. Higher rejections can be achieved if needed by cascading a second filter in the programmable processor.

**Arbitrary RF waveform generation.** We employed the programable photonic processor to generate arbitrary radiofrequency waveforms by generating delayed and scaled replicas of short input optical pulses in the optical domain followed by recombination and scaling to

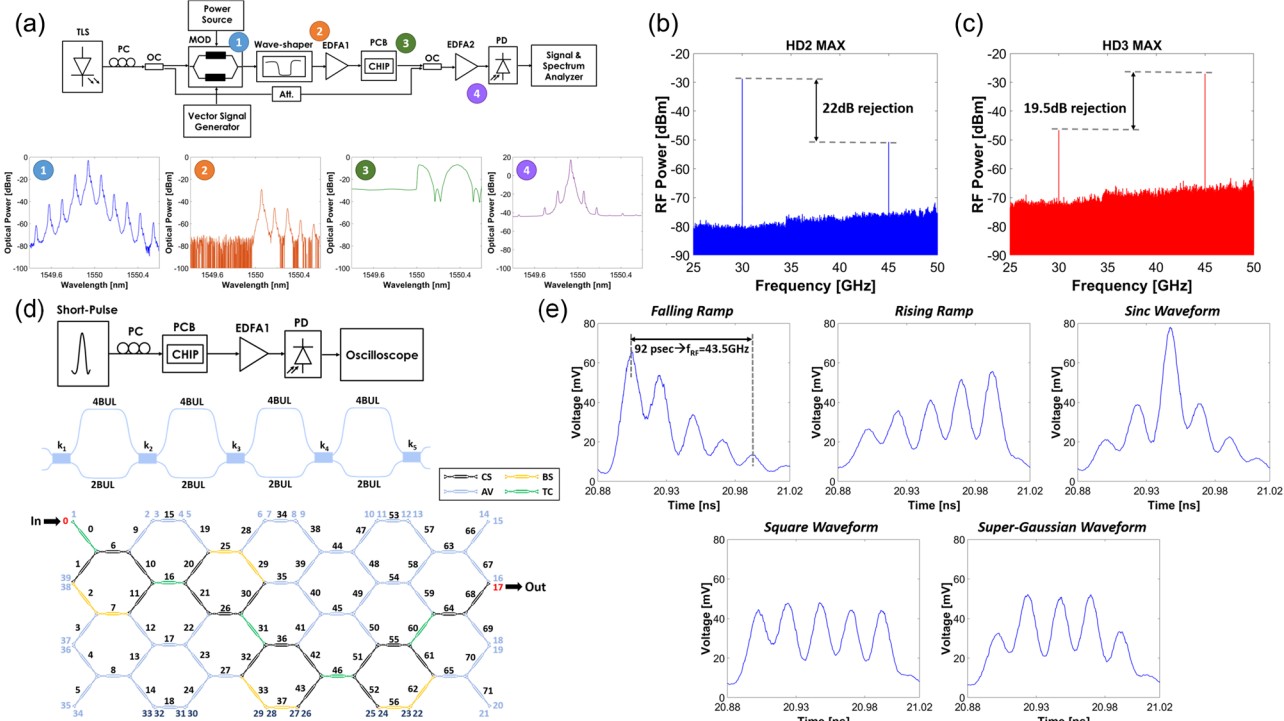

**Fig. 4 | Measured results for optical generation of millimeter-wave CW signals. a** Assembled experimental configuration using the programmable photonics processor. Light from a tunable laser is split into two paths through a 1×2 optical coupler (OC). On one path, the light is delivered to an intensity modulator (MOD) driven by a signal at the RF input frequency (15 GHz) to be multiplied. A tunable optical filter removes both the optical carrier and the lower sidebands at the modulator output. After an amplification stage, a programmable optical integrated filter is synthesized in the photonic processor, to select a single harmonic tone and ensure the suppression of all the remaining spectral components. **b** x2 frequency multiplication (30 GHz) obtained when the third-order UMZI lattice filter shown in Fig. 4d, e is used as a discriminating filter. **c** Same result for the x3 (45 GHz) case. Measured results for arbitrary waveform generation: **d** Experimental setup employed to carry out the AWG functionality by time-domain RF waveform synthesis. A short (4.4 ps) input seed laser pulse (at 5 GHz repetition rate) is processed by a 4th-order UMZI lattice filter (layout shown in the figure) with an arm imbalance of 2BUL. The waveguide mesh programming in the processor is also shown. **e** The processed signal was recorded for different settings of the filter parameters in order to synthesize different waveforms at 43.4 GHz.

achieve different RF wavepackets. These are useful in radar and signal distribution from the central office to base stations. The generation and weighting of the pulse replicas can be implemented by means of a photonic transversal filter. In our case, we chose UMZI lattice filters of different orders (see Supplementary Note 7 for further details). Figure 4d depicts the experimental setup employed to carry out the time-domain RF waveform synthesis, where a short (4.4 ps) input seed laser pulse (at 5 GHz repetition rate) is processed by a 4th-order UMZI lattice filter (layout shown in the figure) with arm imbalance of 2 Basic Unit Lengths. The waveguide mesh programming in the processor is also shown in Fig. 4d. The processed signal was recorded for different settings of the filter parameters in order to synthesize different waveforms. The main results are shown in Fig. 4e. The number of replicas is given in each case by the filter order +1. Time separation between consecutive pulses is around 23 ps which implies an RF central frequency for the resulting wave packet of 43.5 GHz. In Supplementary note 7 we provide more details regarding the possibility of generating RF arbitrary waveforms ranging from the S (3.5 GHz) to the W band (87 GHz).

**Tuneable delay lines and beamforming.** We programmed our processor to enable a four-element beamformer capable of pointing up to 9 angles (4 positive, broadside and 4 negative in a −55° to 55° range where $\Delta L = 1$ BUL→$\theta = 13.7°$, $\Delta L = 2$ BUL→$\theta = 27.4°$, $\Delta L = 3$ BUL→$\theta = 41.25°$, $\Delta L = 4$ BUL→$\theta = 54,9°$ and a similar reversed configuration provided the negative pointing angles. Here, our aim has been to demonstrate the proof of concept of implementing a switched delay line beamformer but indeed having the possibility of pointing more

angles (say 16 or 32 in an aperture with angular span from −60° to 60°) will require scaling-up the waveguide mesh circuit. This result is not far from the maximum reported in the state of the art[49], which is 16 angles (from 0 to 56°) for an 8-element array (around a 3,6° angular step). Furthermore, a similar configuration to ours in terms of radiating elements[58] demonstrates 6 (from −51° to +34° or an angular step of 17°) instead of our 9 pointing angles. The time delay performance and bandwidth of each configuration were measured using a LUNA Optical Vector Analyzer OVA (see details in Supplementary Note 8). Figure 5a shows the measured results for the broadside and positive beam pointing angles for a wavelength of 1550 nm. Each case includes an inset displaying the relative bandwidth around this wavelength. Over 200 GHz bandwidth is easily achieved. The figure also displays the normalized beampattern calculated from the delays at a working Rf frequency of $f = 10$ GHz.

**5 G signal interconnection and power division.** We programmed the processor photonic mesh for the dynamic optical interconnection and power division of 5 G digitally modulated radiofrequency signals. These operations enable more complex tasks like smart signal processing and management in edge data centers or radio-over fiber transmission and 5 G central offices. We first programmed the mesh to perform optical signal interconnection between optical paths with different path lengths and on the other hand, a pair of reconfigurable optical splitters (1×2 and 1x4). Next, we introduced 5 G standard signals (frequency range I 5.9 GHz carrier and Frequency Range II 26 and 37 GHz carriers) to be transmitted through the interconnects and splitters defined in the mesh to evaluate the signal deterioration and

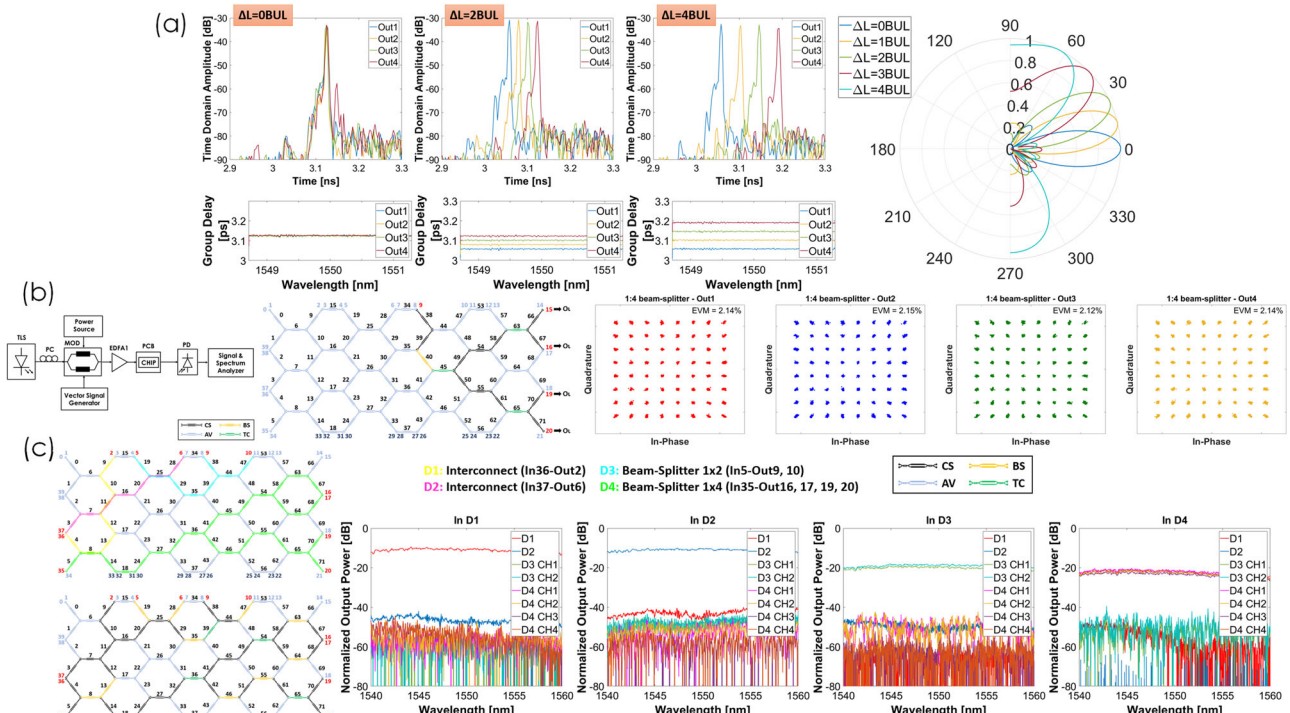

**Fig. 5 | Measured results for Tunable delay lines & Beamforming. a** Measured results for the tunable delay lines implementing the broadside and positive beam pointing angles of a 4-element beamformer for a wavelength of 1550 nm. Insets display the relative bandwidth around this wavelength. The figure also displays the normalized beampattern calculated from the delays at a working RF frequency of $f = 10$ GHz. Measured results for 5 G signal interconnection and power division: **b** Experimental setup, waveguide mesh programming, and constellation results for a 1x4 power divider of a 64-QAM 26 GHz constellation featuring an average EVM of 2.13%, which is substantially lower than the 8% required by the 3GPP TS 38.101-1 standard. Measured results for switching, broadcasting, and multicasting: **c** It shows the results obtained for the switching and broadcasting of broadband signals switching experiment, comprising the selected interconnection pattern (two switching paths between two input and two output paths and two broadcasting connections between one input and two/four output ports), the color diagram for the programmed waveguide mesh and finally the measured results, which were taken with all the connections being simultaneously active and including, for each connection, the effect of undesired crosstalk from the rest.

demonstrate the flexible splitting capability of the mesh. Results (see full detailed description in supplementary note 9) illustrate an excellent performance in terms of error vector magnitude (EVM) for 64 and 256-QAM 5 G constellations. As an example, Fig. 5b shows the details of the experimental setup, waveguide mesh programming, and constellation results for a 1x4 power divider of a 64-QAM 26 GHz constellation featuring an average EVM of 2.13%, which is substantially lower than the 8% required by the 3GPP TS 38.101-1 standard.

**Switching, broadcasting, and multicasting.** We programmed our photonic processor to demonstrate its capabilities to provide switching, broadcasting, and add-drop multiplexing for broadband and channelized signals. Two experiments were carried out addressing the simultaneous implementation of broadband and broadband + narrowband interconnections respectively (see Supplementary Note 10 for a more detailed description). Figure 5c shows the results obtained for the switching and broadcasting of broadband signals experiment, comprising the selected interconnection pattern, which includes two switching (interconnection) paths between two input and two output paths and two broadcasting connections between one input and two/ four output ports, the color diagram for the programmed waveguide mesh and finally the measured results, which were taken with all the connections being simultaneously active and including, for each connection, the effect of undesired crosstalk from the rest. An excellent rejection level of over 33 dB is obtained for switched paths, while broadcast signals provide a minimum rejection close to 22 dB. With larger waveguide meshes and further optimization of routing paths, it is expected that these already remarkable results can be further improved[59]. Note that a 40 nm operation range is obtained.

**Frequency measurement.** Modern electronic systems use advanced receivers to detect and identify signals. They often start with Instantaneous Frequency Measurement (IFM) receivers, which, while common and high-performing, are typically limited to around 20 GHz due to electronic circuit bandwidth constraints. A photonic-assisted technique consists of using an RF-photonic filter with an optical complementary filter to obtain an Amplitude Comparison Function (ACF) that has a monotonic relationship with the RF frequency within a certain range. We programmed the waveguide mesh in our photonic processor as shown in Fig. 6a to implement a 1 input 2 complementary UMZI (2 BUL) filter outputs (see Fig. 6b required for the computation of the ACF of a frequency measurement system. An ACF shown in Fig. 6c features a maximum 2.77 dB/GHz slope in a spectral range of >15 GHz. Further details are provided in Supplementary Note 11.

**Optoelectronic oscillation.** Finally, we programmed our photonic processor to demonstrate its capabilities to implement optoelectronic oscillators with reconfigurable capabilities. In the first experiment, we exploited internal cavity switching to change from a short to a long cavity device. Figure 6d shows the layout for the first experiment, where two cavities (paths 1 and 2) are switched using our programmable processor as shown in the upper part of the Figure. The short cavity OEO (path 1) features an optical path of 53 m and an electronic path of 1.35 m (overall spectral period around 3.1 MHz). The long cavity OEO (path 2) features an optical path of 1.4 Km and an electronic path of 1.35 m (overall spectral period around 1.38 kHz). In each case, we measured the OEO spectrum around 1.5 GHz and 10 GHz by inserting a 50 and a 430 MHZ filter respectively centered at those frequencies. The main results are shown in Fig. 6d. In a second experiment, detailed

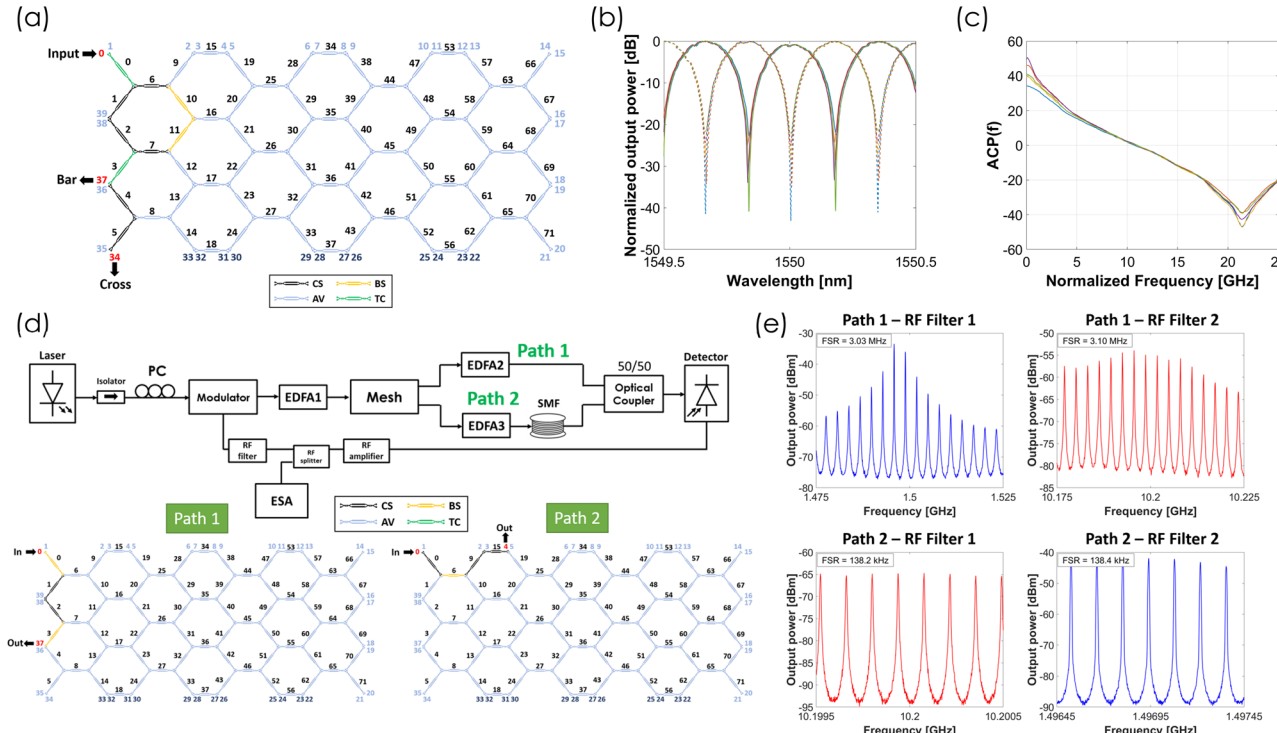

**Fig. 6 | Measured results for Frequency measurement and Optoelectronic oscillator. a** Color diagram for the programmed waveguide mesh used to implement a 1×2 MZI Filter with path imbalance given by 2BUL (Green: Tunable Coupler, Black: Cross State, Ochre: Bar state, Blue: unused). **b** Measured results for the 1 × 2 MZI filter showing Bar (even maxima) and Cross (odd maxima) transfer functions and different values of the MZI coupling constants ranging from 0.5 to 0.58. **c** Measured ACF (f) for the transfer functions displayed in Fig. 6b and zero extra phase shift between the arms. **d** Layout of the experimental configuration assembled to demonstrate a switched OEO configuration by suitable programming of the RF-Photonic processor. The upper part shows the internal programming of the waveguide mesh to implement the short (path 1) and the long (path 2) OEOs. **e** The spectra of both OEOs around 1.5 and 10 GHz.

in supplementary note 12, we programmed the same processor to implement via a power division operation a dual loop Vernier OEO and increase the overall cavity-free spectral range.

## Discussion

We have described the implementation of a general-purpose photonic processor that relies on a technology stack comprised of a photonic processing hardware layer, an electronic monitoring, driving, and control layer, and a software layer that enables device programming and resource optimization. We have demonstrated that this device can implement all the main functionalities required in Microwave Photonic Systems. Prior demonstrations using application-specific integrated photonic chips have been limited to the implementation of one functionality and with little degree of reconfigurability and flexibility. The proposed device overcomes these limitations by providing full flexibility in terms of functionality selection as well as in terms of parameter reconfiguration for the selected application. Furthermore, we have demonstrated that simultaneous implementation of several functionalities in the photonic core is possible and therefore that parallel processing operations are feasible. Operation frequency ranges in the 15 to 45 GHz band have been demonstrated but even higher frequency ranges can be achieved by reducing the BUL. We estimate that the current value of 811 μm can be lowered to around 200 μm thus reaching beyond 200 GHz operation bandwidth. Additionally, to narrow down the gap between application-specific circuits and programmable devices, better quality filters with increased contrast ratios would demand PUC loss reduction. Our current value of around 0.48 dB/PUC can be lowered to figures around 0.1 dB/PUC and moreover, overall loss compensation may be possible by means of the integration of optical amplification at selected points of the silicon photonics core leveraging micro-transfer printing techniques[60].

Recent work[61] has outlined the optimum placement of such amplifiers in a programmable photonic mesh core. Both loss and BUL reduction will lower the scalability barriers for integrating more PUCs in the photonic core opening the path to the implementation of more complex architectures. We estimate that the current figure of 1.91 actuators per mm$^2$ chip can be upgraded to 10 actuators per mm$^2$ in the near-middle term with state-of-the-art foundry processes and advanced flip-chip packaging. Furthermore, with a power consumption of around 1–2 mW/π per phase shifter already achievable, we envisage full cores with over one thousand operating PUCs consuming 1 watts or less. The on-chip incorporation of optimized high-performance blocks such as RF modulators, detectors, amplifiers, and specialty filters will enable the implementation of the functionalities described here with improved figures of merit.

An interesting point of discussion is related to the comparison between results that can be achieved with FPPGA and ASPIC approaches in terms of performance and for each functionality. This study is still missing in the literature, perhaps due to the early stage of development of both approaches, but a fair comparison would need to consider many aspects such as power consumption, reconfigurability speed, insertion losses, and footprint scalability. Although this analysis is beyond the scope of this paper, we would like to highlight that key parameters will be heavily dependent on the achievable waveguide propagation and scattering losses, the power consumption of phase shifting mechanisms, and their tuning speed. It is to be expected that as it happens in electronics ASPICs will provide a much better solution for very focused applications and high production volumes.

Our approach leveraging the resource reusability provided by a general-purpose architecture will bring key benefits in terms of lower costs by reducing non-recurrent engineering costs, leveraging economies of scale in fabrication, increased sustainability by the

drastic reduction in resources from unsuccessful fabrication runs, and lower power consumption. We have seen these approaches succeeding before in the electronics arena, they are best known as field programmable gate arrays (FPGAs) and microprocessors. In photonics, this disruptive concept is called to play a similar role in future years and extend its application to a broad category of fields such as photonic computing, advanced communications, sensors, lidar, and spectroscopy.

## Methods

### Device manufacturing and assembly process

The photonic core was fabricated using 130 nm lithography process in SOI wafers with a 220-nm thick silicon overlayer and a 3-μm thick buried oxide layer. The process allows the doping of waveguides and under-etched waveguides to improve the efficiency of the phase actuators and mitigate thermal crosstalk effects. Germanium on silicon is employed for on-chip photodetection. These dies were then mounted onto a copper chuck employing thermal epoxy. A Printed Circuit Board (PCB) is also attached to the copper structure and a wire bonding process was used to provide electrical connections between the die and the PCB. A fiber array with a pitch distance of 127 μm was fixed to the on-chip edge coupler array of the die by active alignment and epoxy.

### Device testing process

Manufacturing variations and thermal gradients affect the optical phase of the elements, making it a challenge to guarantee the robustness of the functionalities that the processor provides[62]. We performed a static characterization of the passive building blocks employing the test cells to extract information regarding the main optical properties. To get the spectral responses, we have employed a tunable laser (ANDO AQ4321D) featuring a 1 pm wavelength resolution and an optical spectrum analyzer (ANDO AQ6217C). For the electro-optical characterization of the PUCs, we employed external high-precision current sources (Keihtley2401) and a custom-made programmable array described as a driving unit in the main test. The calibration of the mesh was completed through an automated process that relies on graph modeling. This process was carried out for all the thermal tuners present on the circuits and HPBs. The calibration process was run once at 15°, 25°, and 35° to validate the robustness of the result vs temperature variations. See details of the algorithm in Supplementary Note 2.

Additionally, we tested the robustness of the actuators with more than 10,000 continuous 0–3π cycles showing no component deterioration (change in the resistance).

For time-domain measurements, which were crucial in characterizing the time delay response of the mesh when programmed to implement true time delay lines, tunable delay lines, and beamforming, we relied on an optical vector analyzer (LUNA).

Conversely, when it came to characterizing the functionalities related to Microwave Photonics (MWP), we set up various configurations, each requiring distinct equipment (for detailed descriptions, please refer to Supplementary Notes 4 to 12).

In the initial stage of light generation for all MWP applications, we consistently used a tunable laser source (TUNICS T100R/M YENISTA), except for the generation of arbitrary RF waveforms, where a 10 GHz high-speed fiber laser from Calmar Optcom emitting picosecond short pulses was employed.

In several MWP functionalities, modulation of the light signal was necessary, such as in the implementation of tunable and reconfigurable RF filters, optical generation of mm-wave CW signals, optoelectronic oscillators, and the demonstration of 5 G signal interconnection and power division. To accomplish this modulation, we employed an electro-optical dual-drive modulator (Sumitomo T.DEH1.5-40-ADC). Additionally, for generating the

modulating signals to modulate the optical carrier, we utilized an optical network analyzer (Agilent N4373C) for tunable and reconfigurable RF filter implementation, and a vector signal generator (Rohde and Schwarz SMW200A) for the optical generation of mm-wave CW signals or 5 G signal interconnection and power division applications, to generate the required digital modulated signals. To accomplish detection, we employed a high-speed optical photodetector (Finisar XPDV3120R-VF-VA). In most cases, amplification stages were necessary, for which we utilized erbium-doped fiber amplifiers (Amonics AEDFA-23-B-FA and AEDFA-27-B-FA). Additionally, in specific experiments where signal filtering was required, such as in the characterization of tunable and reconfigurable RF filters or the optical generation of mm-wave CW signals for carrier suppression and single-sideband modulation, we employed a tunable optical filter (Finisar WaveShaper 4000 s).

Finally, depending on the specific application, various analyzers were used after photodetection. For instance, we employed a vector network analyzer (Agilent N4373C) to obtain electrical responses for tunable and reconfigurable RF filters. For the optical generation of mm-wave CW signals, we used a signal and spectrum analyzer (Rohde and Schwarz FSW43) to visualize the generated mm-wave tones, measure their phase noise, or evaluate the error vector magnitude of received constellations in the 5 G signal interconnection and power division experiment. In the case of arbitrary RF waveform generation, the generated waveform was captured using an optical sampling oscilloscope (TEKTRONIX DSA8200), and oscillations were measured when implementing optoelectronic oscillators using a signal analyzer (Agilent N9020A).

## Data availability

All data are available from the corresponding author upon request.

## Code availability

All codes are available from the corresponding author upon request.

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

## Acknowledgements

D.P.L. acknowledges funding through the ERC-StG-2022-LS-Photonics-101076175. J.C. acknowledges funding from the ERC Advanced Grant ERC-ADG-2016-741415 UMWP-Chip and ERC-POC-2019-959927, Generalitat Valenciana funding through Future MWP technologies and applications PROMETEO/2017/103, Advanced Instrumentation for World Class Microwave Photonics Research IDIFEDER/2018/031, EUIMWP

CA16220, Infraestructura para caracterización de Chips Fotónicos EQC2018-004683-P. All the authors also acknowledge funding from HORIZON-EIC-2021-TRANSITION OPEN-01101057934 INSPIRE.

## Author contributions

D.P.L., P.D., and J.C. conceived the processor's high-level design. D.S., M.G. A.L.H., J.F., D.P.G., and D.P.L. designed the chip and high-performance blocks. A.G., D.P.L., and J.C. conceived the experiments. A.G. executed the measurements. D.S. M.G. A.L.H A.C, Z.X, E.S.G., and D.P.L. conceived and developed the software layer. N.B., J.B., J.F., and DPL lead the assembly and thermomechanical works. A.G. A.S. assisted with the component-level validations. A.Q., A.C, A.S assisted with the control layer development. J.C., A.G., A.M., and D.P.L. analyzed the data and wrote the paper. D.P.L. and J.C. managed, coordinated, and supervised the project.

## Competing interests

The authors declare no competing interests.
