## [Peer Review File · Nature Communications]

General purpose programmable photonic processor for advanced radiofrequency applicationsREVIEWER COMMENTS

Reviewer #1 (Remarks to the Author):

The authors propose a programmable photonic circuit comprising the optical layer, the control layer and the software layer. The multifunctional circuit can implement all the main functionalities required in microwave photonic systems. This work measures the loss and true time delay, as well as effectively demonstrates functionalities of optical filters, RF filters, RF phase shifters, arbitrary RF waveform generation, tunable delay lines, tunable RF beamforming, 5G signal interconnection and power division, switching, broadcasting, add-drop multiplexing, and so on, indicating the capability of performing a programmable photonic processor with the capability to implement so many functionalities.

1) The programmable photonic processor has been presented in the reference [1], and this work provides more detailed instructions of the processor, and also implements the tests. I would consider this work is expanded to consider its functionalities in microwave photonic systems, any improvements to the processor core must be clarified.

2) According to the literature "Xu X., Ren G., Feleppa T., et al. Self-calibrating programmable photonic integrated circuits. Nat. Photon. 16, 595-602 (2022).", due to manufacturing variations and thermal gradients that affect the optical phase of the elements, it is difficult to guarantee the robustness of the functionalities that the processor provides. Would the processor need to implement self-calibrating, or has the long-term stability of the circuits been tested as well?

Reviewer #2 (Remarks to the Author):

In this manuscript, the authors present the first complete working prototype of a field programmable photonic gate array (FPPGA), including the control electronics and software layer for programming and optimization. This is an impressive technological feat that represents the culmination of many years of work by the authors, as testified by their previous publications. The authors then demonstrate the potential of their FPPGA potential by implementing many systems commonly used to process radio frequency (RF) signals optically. This work defines the state-of-the-art in FPPGA and thus will be of interest to the integrated photonic community. Nevertheless, I have questions and suggestions for the authors which I hope will help improve the manuscript.

1.The authors say that (lines 85-87) FPPGAs could provide huge gains "in terms of ultra-high bandwidth, high-speed operation, flexibility, low power consumption, and reduced fabrication costs from leveraging economies of scale." Are the authors claiming that FPPGAs could out perform application-specific photonic integrated circuits (ASPICS) in bandwidth, speed and power consumption? If so, could they explain their rational in more details?

2.The result section presents a long list of systems implemented using the FPPGA. However, it would be informative to very briefly compare the results obtained with the FPPGA with those of recent demonstrations using ASPICS to show if there is a significant difference in performance between the two approaches.

3.Could the filtering functions implemented with the Wave-shaper in the test setups used to demonstrate RF filters and the generation of MM-wave CW signals be realized with the FPPGA instead? If so, why did you choose to use this configuration for the measurement setups?

4.Could the authors comment on the practicality of having a beamforming system that can scan 9 angles? What would be necessary to significantly improve the resolution? Could a small range of

analog tuning be achieved by controlling the phase inside the basic units?

5. The functions implemented by the FPPGA in the optoelectronic oscillator experiments are very simple. Are there significant benefits in using a FPPGA in this case? Could you have used the on-chip photodetectors to maximize the use of the FPPGA? What is the bandwidth of those photodetectors?

The following comments are minor formatting suggestions.

On line 41, I would say the ever-increasing demands for bandwidth instead of demands of bandwidths.

There is a space missing on line 130 before thermo-optic.

Line 160 should end with inclusively, not inclusive.

In figure 2(b), and in all the other schematics showing the photonic mesh (e.g., figs. 3(a), 3(b), 4(d), 5(b), 5(c), 6(a) and 6(b)), it is very difficult to see the numbers and identify the inputs and outputs.

The color code identifying the functionality of each basic element in the mesh should be presented in fig. 3 where it is used for the first time.

The legend and the different curves in figs. 3(b) right, 3(c) right, 5(c) are very hard to see. This also applies to fig. S9.2(c) in the supplementary notes.

The caption of fig. 3 is missing a description for parts d, e and f.

The text on line 225 appears to be referring to fig. 3(d)-(e), not fig. 4.

Figs. 6(b) and 6(c) should have a legend to identify the different curves. This also applies to figs. S3.5, S3.9, S3.12(a), S3.12(b), S3.14 in the supplementary notes.

On lines 303-305, the short and long cavities appear to have the same length according to the description.

On page 14 of the supplementary notes, there are two different extinction ratios given for the reflection case in the description of the ring resonators. I presume that one of them refers to the transmitted signal.

The words there is a are repeated twice in a row in the last paragraph of page 37 of the supplementary notes.

In conclusion, I believe that the manuscript will be suitable for publication after minor modifications.

Best regards,

Michaël Ménard
Associate Professor
École de technologie supérieure

REVIEWER COMMENTS

Reviewer #1 (Remarks to the Author):

The authors propose a programmable photonic circuit comprising the optical layer, the control layer and the software layer. The multifunctional circuit can implement all the main functionalities required in microwave photonic systems. This work measures the loss and true time delay, as well as effectively demonstrates functionalities of optical filters, RF filters, RF phase shifters, arbitrary RF waveform generation, tunable delay lines, tunable RF beamforming, 5G signal interconnection and power division, switching, broadcasting, add-drop multiplexing, and so on, indicating the capability of performing a programmable photonic processor with the capability to implement so many functionalities.

1) The programmable photonic processor has been presented in the reference [1], and this work provides more detailed instructions of the processor, and also implements the tests. I would consider this work is expanded to consider its functionalities in microwave photonic systems, any improvements to the processor core must be clarified.

Reply: We thank the reviewer for this appreciation. Reference [1] is a paper on programmable photonic circuits in general. This paper reviews several topics including the underlying photonic programmable waveguide mesh architectures, the technology stack, and, of course, the architecture of the general-purpose photonic processor (which by the way was originally proposed in [2] and [3]). It also covers very briefly the salient applications fields and, in particular, Microwave Photonics.

As it is written there.... Photonic circuits for microwave applications are mostly ASPICs implementing filtering, waveform generation, reconfigurable delay lines, or frequency measurements. These functions can all be implemented in generic recirculating waveguide meshes.....

Thus in [1] we anticipated the possibility of implementing all these functionalities in a general-purpose programmable processor. This demonstration is contingent on developing a suitable technology stack (photonics, electronics, and software) and this is the first time that this has been reported.

Action taken: we have included the following paragraph:

So far, individual microwave photonic applications are mostly carried by ASPICs, but as anticipated¹, all of them can be implemented by the same general-purpose programmable processor if it integrates the required technology stack (photonic, electronic and software). This evolution beyond the current state of the art and the demonstration of multifunctionality by programming are the fundamental results of the paper.

2) According to the literature “Xu X., Ren G., Feleppa T., et al. Self-calibrating programmable photonic integrated circuits. Nat. Photon. 16, 595-602 (2022).”, due to manufacturing variations and thermal gradients that affect the optical phase of the elements, it is difficult to guarantee the robustness of the functionalities that the processor provides. Would the processor need to implement self-calibrating, or has the long-term stability of the circuits been tested as well?

Reply: Thanks for your question. As mentioned by the reviewer, it is well known that interferometric photonic components are very sensitive to design and fabrication errors given by conventional fab processes and wafer geometry variations. Even nano-meter scale deviations of a waveguide geometry can cause random phase shifts when propagated over a tens of micrometer length. The first consequence for a large-scale design as the presented in this work is that the initial (passive) status of the interferometers included in the design is random rather than cross-state.

As covered in the methods section: *The calibration of the mesh was completed through an automated process that relies on graph modeling. This process was carried out for all the thermal tuners present on the circuits and HPBs.*

In supplementary note 1 we highlight the reference that includes the calibration algorithm employed:

Programmable photonics involves a software framework that encompasses a wide array of algorithms, methods, or routines executed by an electronic processing unit. These serve primarily to configure specific functionalities within a Photonic Integrated Circuit (PIC) and to optimize the backplane.

*These routines can be categorized based on their intended outcomes or their specific requirements. **Among them, a group of fundamental routines is dedicated to calibrating and pre-characterizing the photonic integrated circuit. For instance, iterative routines can be employed to calibrate the electro-optical response of each phase shifter within the circuit, eliminating the need for integrating optoelectronic monitors into the waveguide mesh arrangement [1].** Additionally, these routines can extract valuable information such as the power consumption of individual units, accumulated losses, and identifying and recording defects within the circuit.*

The data collected through these routines can then be utilized by a set of algorithms designed to optimize both the circuit's functionality and resource utilization. One such algorithm is the auto-routing algorithm [2].

[1] A. López-Hernández, M. Gutiérrez-Zubillaga and D. Pérez-López, "Automatic Self-calibration of Programmable Photonic Processors," 2022 IEEE Photonics Conference (IPC), Vancouver, BC, Canada, 2022, pp. 1-2, doi: 10.1109/IPC53466.2022.9975587.

On the other hand, temperature gradients are susceptible of changing the circuit performance during operation. First, we mitigate this issue at the design level with an efficient phase actuator design based on under-etched waveguides. This reduces the power dissipated in the actuator and increase the isolation with neighbouring waveguides. Secondly, as described in the methods, the *dies were then mounted onto a copper chuck employing thermal epoxy*. This reduces the thermal barrier in the thermal control loop. Finally, we validated and characterized the thermal crosstalk and obtained that residual thermal crosstalk is kept in the die, being non-dominant when compared to other sources of crosstalk (electrical, optical, etc). In any case, the crosstalk is always kept better than 25 dB, being 30 dB a typical figure.

As a final consideration, we validated that the calibration curves remained valid both for a $\pm 10^\circ\text{C}$ temperature variation and for a long-term period of non-continuous usage (currently more than 6 months). Test variations of 10000 continuous cycles over a phase shift have been validated without component deterioration (change in the resistance).

Action taken:

We have emphasized this in the methods including the following statements and a new reference

Manufacturing variations and thermal gradients affect the optical phase of the elements, making it a challenge to guarantee the robustness of the functionalities that the processor provides⁶⁴.

(...)

The process allows the doping of waveguides and under-etched waveguides to improve the efficiency of the phase actuators and mitigate thermal crosstalk effects.

(...)

The calibration of the mesh was completed through an automated process that relies on graph modeling. This process was carried out for all the thermal tuners present on the circuits and HPBs. The calibration process was run once at 15°, 25° and at 35° to validate the robustness of the result vs temperature variations. See details of the algorithm in Supplementary Note 1.

Additionally, we tested the robustness of the actuators with more than 10000 continuous $0-3\pi$ cycles showing no component deterioration (change in the resistance).

Reviewer #2 (Remarks to the Author):

In this manuscript, the authors present the first complete working prototype of a field programmable photonic gate array (FPPGA), including the control electronics and software layer for programming and optimization. This is an impressive technological feat that represents the culmination of many years of work by the authors, as testified by their previous publications. The authors then demonstrate the potential of their FPPGA potential by implementing many systems commonly used to process radio frequency (RF) signals optically. This work defines the state-of-the-art in FPPGA and thus will be of interest to the integrated photonic community. Nevertheless, I have questions and suggestions for the authors which I hope will help improve the manuscript.

Reply: We appreciate very much the encouraging words by the reviewer and agree that his positive comments will help to improve the paper.

1.The authors say that (lines 85-87) FPPGAs could provide huge gains “in terms of ultra-high bandwidth, high-speed operation, flexibility, low power consumption, and reduced fabrication costs from leveraging economies of scale.” Are the authors claiming that FPPGAs could outperform application-specific photonic integrated circuits (ASPICS) in bandwidth, speed and power consumption? If so, could they explain their rational in more details?

Reply: We thank the reviewer for raising this point. No, and we apologize for the misunderstanding. We refer to the cases where programmable photonic circuits can replace electronic circuits. There, the gains in terms of bandwidth and power consumption when moving high-speed data are notorious. The advantages of FPPGAs as compared to ASPICs are outlined in some paragraphs before.

Action taken: We have rewritten the paragraph to make this point clear. It now reads:

(...) While to the best of our knowledge, this has not been demonstrated so far, the gains that could be achieved in the cases where programmable photonic circuits can replace electronic subsystems are huge, especially in terms of ultra-high bandwidth, high-speed operation and low power consumption. These are complemented by their flexibility and reduced fabrication costs from leveraging economies of scale.

2.The result section presents a long list of systems implemented using the FPPGA. However, it would be informative to very briefly compare the results obtained with the FPPGA with those of recent demonstrations using ASPICS to show if there is a significant difference in performance between the two approaches.

Reply: We thank the reviewer for this comment. Indeed having the possibility of establishing fair comparisons with ASPIC-based results is of great interest. The main difficulty here is that ASPICs are reported in a myriad of configurations, and also in different material platforms so, for example, in the case of filters, quality factors are usually superior in ring-based ASPICs built on silicon nitride as losses are considerably lower than those of SOI. In the case of delay lines longer paths can again be achieved using switched spiral silicon nitride (SiN) waveguides but then the footprint is much higher due to the lower refractive index contrast. Phase shifters on the contrary are usually more efficient in SOI than in SiN, at least for the case of thermo-optic tuning, while both of them are slow if compared to structures based on InP, which are fast but

lossy. In addition, not all the functionalities have been completely demonstrated in ASPICs. For example, integrated OEOs have been reported, where part of the circuit is an ASPIC and part is a discrete electronic board. This means that even if we can show and compare bold figures we feel that it may make little sense at this stage of development.

Although a really fair comparison would be very hard to list in the paper we nevertheless want at least to briefly address the issue

Action taken: We have included a paragraph in the conclusions section highlighting where the main performance differences between ASPICs and FPPGA-enabled functionalities are expected to come from.

An interesting point of discussion is related to the comparison between results that can be achieved with FPPGA and ASPIC approaches in terms of performance and for each functionality. This study is still missing in the literature, perhaps due to the early stage of development of both approaches, but a fair comparison would need to consider many aspects such as power consumption, reconfigurability speed, insertion losses, and footprint scalability. Although this analysis is beyond the scope of this paper, we would like to highlight that key parameters will be heavily dependent on the achievable waveguide propagation and scattering losses, the power consumption of phase shifting mechanisms, and their tuning speed. It is to be expected that as it happens in electronics ASPICs will provide a much better solution for very focused applications and high production volumes.

3. Could the filtering functions implemented with the Wave-shaper in the test setups used to demonstrate RF filters and the generation of MM-wave CW signals be realized with the FPPGA instead? If so, why did you choose to use this configuration for the measurement setups?

Reply: Thank you very much for your comment. The version of our photonic chip did not integrate the external modulators. Thus, for our experiments, we needed to resort to a discrete dual-drive external modulator, which produces double sideband modulation (in linear operation) and harmonics (in nonlinear operation). We employed the waveshaper filter to suppress a sideband in the RF photonic filter experiments as well as the carrier and the undesired sideband in the MM-WW signal generation experiments. The operation of the waveshaper is purely passive, so the answer is yes, we could have implemented a filter in the waveguide mesh for that purpose. We chose however to use the waveshaper for convenience because we needed a strong suppression of the undesired input signals to the programmed filters and this would have implied (due to the PUC insertion losses) the need for cascading stages, consuming valuable area within the mesh which we needed to implement the layouts required for the functionalities. This limitation will be solved in future evolutions of the chip since there is considerable room for technological improvement to scale them up. In particular, loss reduction per PUC and PUC size are essential for chip scalability. For instance, using new coupler designs can readily reduce the 0.49dB/PUC losses per PUC to 0.1dB/PUC. Also, the footprint (length) of our PUCs can be reduced by a factor of 3. We also expect that future versions of the chip will integrate the dual-drive modulator driven by a hybrid.

4. Could the authors comment on the practicality of having a beamforming system that can scan 9 angles? What would be necessary to significantly improve the resolution?

Could a small range of analog tuning be achieved by controlling the phase inside the basic units?

Reply: Thank you very much for the comment. For sure a beamformer with the ability of scanning 9 angles is not very practical although it depends on the particular application and the required resolution. Here our aim has been to demonstrate the proof of concept of implementing a switched delay line beamformer but indeed that having the possibility of pointing more angles (say 16 or 32 in an aperture with angular span from -60° to 60°) will require scaling-up the waveguide mesh circuit. The following table summarizes some recent state of the art results published in the literature for switched delay line integrated Microwave Photonic Beamformers

Technical Approach/ Reference	Structure	Platform	Footprint (mm ²)	Bandwidth (GHz)	Delay range (ps)	Power consumption (mW)	Beam Angle range (°)	Angles	Channel number
Switched Delay Lines [R1]	MZI-TTODLs	SOI	11.03 x 3.88	10(*)	0 - 496	1450	0 to 75.64	16	8
Switched Delay Lines [R2]	MZI-TTODLs	SiN	32 x 8	NS	0 - 22.5	NS	-51 to 34	6	4
Switched Delay Lines [R3]	MZI-TTODLs	SOI	7.4 x 1.8	NS	0 - 191.4	1251	-	-	1

(*) Only for positive beam steering angles

[R1] Zhu C., Lu L., Shan W., Xu W., Zhou G., Zhou L., & Chen J. Silicon integrated microwave photonic beamformer. *Optica*. 7, 1162–1170 (2020).

[R2] Liu Y., Isaac B., Kalkavage J., Adles E., Clark T., & Klamkin J. 93- GHz signal beam steering with true time delayed integrated optical beamforming network. *Optical Fiber Communication Conference (OFC)*. (2019).

[R3] Zheng P., Wang C., Xu X., Li J., Lin D., Hu G., Zhang R., Yun B., & Cui Y. A seven bit silicon optical true time delay line for ka-band phased array antenna. *IEEE Photonics Journal*. 11, 4 (2019).

As it can be seen, the maximum demonstrated is 16 angles (from 0 to 56°) for an 8-element array (around a $3,6^\circ$ angular step). For instance, a similar configuration to ours in terms of radiating elements [R2] demonstrates 6 (from -51° to $+34^\circ$ or an angular step of 17°) instead of our 9 pointing angles. We agree with the reviewer that this figure needs to be scaled to render this as a practical technology solution. To increase resolution two actions are needed. First, increase the number of radiators as the 3dB beamwidth is inversely proportional to this figure and second, to increase the number of pointing angles to the angle step is compatible with the 3 dB resolution. This last figure depends on the number of independent incremental delay paths that can be established in the mesh. We are confident that future waveguide mesh designs where the number of PUCs will be scaled up to a figure of x4 will allow us to achieve in between 16 (4 bit) and 32 (5 bit) pointing angles.

Action taken: We have introduced the following text in the main manuscript:

Here our aim has been to demonstrate the proof of concept of implementing a switched delay line beamformer but indeed that having the possibility of pointing more angles (say

16 or 32 in an aperture with angular span from -60° to 60°) will require scaling-up the waveguide mesh circuit. This result is not far from the maximum reported in the state of the art⁵⁹, which is 16 angles (from 0 to 56°) for an 8-element array (around a $3,6^\circ$ angular step). Furthermore, a similar configuration to ours⁶⁰ in terms of radiating elements demonstrates 6 (from -51° to $+34^\circ$ or an angular step of 17°) instead of our 9 pointing angles.

And we included the references:

[59] Zhu C., Lu L., Shan W., Xu W., Zhou G., Zhou L., & Chen J. Silicon integrated microwave photonic beamformer. *Optica*. 7, 1162–1170 (2020).

[60] Liu Y., Isaac B., Kalkavage J., Adles E., Clark T., & Klamkin J. 93- GHz signal beam steering with true time delayed integrated optical beamforming network. *Optical Fiber Communication Conference (OFC)*. (2019).

5.The functions implemented by the FPPGA in the optoelectronic oscillator experiments are very simple. Are there significant benefits in using a FPPGA in this case? Could you have used the on-chip photodetectors to maximize the use of the FPPGA? What is the bandwidth of those photodetectors?

Reply: Thank you very much for your comment. Yes indeed the experiments were very simple. In this case we were again limited by the fact that the external modulator was not integrated in the chip. The ideal configuration would have been to connect the integrated photodiode output to the RF modulator input to have a compact cavity and then switch inside the mesh different delay line and interconnection paths to demonstrate OEOs with different spectral configurations and multi-GHz spectral periodicity. Since this was not possible, we decided to use the mesh to switch and combine different external cavities implemented using fibre links. This is the reason why, for this experiment, we employed an external photodiode. Though these are not the ideal experiments we wanted to carry they are nevertheless of interest as we have been able to show cavity switching and Vernier effect. As stated above in point 3, we also expect that future versions of the chip will integrate the external modulator and then demonstrate more interesting results.

The following comments are minor formatting suggestions.

On line 41, I would say the ever-increasing demands for bandwidth instead of demands of bandwidths.

Action taken: corrected as suggested

There is a space missing on line 130 before thermo-optic.

Action taken: corrected

Line 160 should end with inclusively, not inclusive.

Action taken: corrected as suggested

In figure 2(b), and in all the other schematics showing the photonic mesh (e.g., figs. 3(a), 3(b), 4(d), 5(b), 5(c), 6(a) and 6(b)), it is very difficult to see the numbers and identify the inputs and outputs.

Action taken: We have modified the figures expanding the port numbers in the mesh diagrams to make them more legible.

The color code identifying the functionality of each basic element in the mesh should be presented in fig. 3 where it is used for the first time.

Action taken: Color code has been introduced

The legend and the different curves in figs. 3(b) right, 3(c) right, 5(c) are very hard to see. This also applies to fig. S9.2(c) in the supplementary notes.

Action taken: We have modified the legends on these curves to make them more legible.

The caption of fig. 3 is missing a description for parts d, e and f.

Action taken: We have included the missing description in the caption.

The text on line 225 appears to be referring to fig. 3(d)-(e), not fig. 4.

Action taken: Correct!, we have changed the reference to figures 3(d)-(e).

Figs. 6(b) and 6(c) should have a legend to identify the different curves. This also applies to figs. S3.5, S3.9, S3.12(a), S3.12(b), S3.14 in the supplementary notes.

Action taken: It is hard to introduce the legends because of space restrictions. In figures 6(b) and (c) we only wanted to convey the message that the variation of the transfer functions within a selected range of coupling constants is very small. Thus a reference on the range covered is inserted in the figure caption. *(b) Measured results for the 1x2 MZI filter showing Bar (even maxima) and Cross (odd maxima) transfer functions and different values of the MZI coupling constants ranging from 0.5 to 0.58.*

As for the supplementary material figures, they correspond to complex filter settings involving for each case different values for different PUCs. We cannot include all the values for practical space reasons but even if we could these would not be very informative as the real impact is on the location of the filter poles and zeros. Our aim is to show that the pole and zero locations can be varied and therefore the filter transfer function shapes and extinction ratios changed.

On lines 303-305, the short and long cavities appear to have the same length according to the description.

Action taken: This is correct, there was a mistake. The correct figures have been added:

The short cavity OEO (path 1) features an optical path of 53m and an electronic path of 1.35 m (overall spectral period around 3.1 MHz). The long cavity OEO (path 2) features an optical path of 1.4 Km and an electronic path of 1.35 m (overall spectral period around 1.38 kHz).

On page 14 of the supplementary notes, there are two different extinction ratios given for the reflection case in the description of the ring resonators. I presume that one of them refers to the transmitted signal.

Action taken: This was a mistake. Indeed the second one refers to the transmission case. It now reads:

Moreover, we chose coupling factors for a maximum extinction ratio of around 25dB in the reflection case and around 10dB in the transmission case

The words there is a are repeated twice in a row in the last paragraph of page 37 of the supplementary notes.

Action taken: corrected

In conclusion, I believe that the manuscript will be suitable for publication after minor modifications.

Best regards,

Michaël Ménard
Associate Professor
École de technologie supérieure

REVIEWERS' COMMENTS

Reviewer #1 (Remarks to the Author):

The authors have provided some modifications in the revised version to my concerns, however, Compared to the work in the reference [1], I am not convinced, as the authors claimed, that this is the first time that a suitable technology stack (photonics, electronics, and software) is reported, as these two are quite similar in Ref. [1].

Reviewer #2 (Remarks to the Author):

I would like to thank the authors for taking the time to thoroughly answer my questions. I believe that the manuscript is now ready for publication.

Best regards,

Michaël Ménard
Associate Professor
École de technologie supérieure

Reviewer #1 (Remarks to the Author): The authors have provided some modifications in the revised version to my concerns, however, Compared to the work in the reference [1], I am not convinced, as the authors claimed, that this is the first time that a suitable technology stack (photonics, electronics, and software) is reported, as these two are quite similar in Ref. [1].

We have removed this statement from the abstract although we believe that it is the first time that is demonstrated.